# The Significance of Blood and Peritoneal Fluid Biochemical Markers in Identifying Early Anastomotic Leak following Colorectal Resection—Findings from a Single-Center Study

**DOI:** 10.3390/medicina58091253

**Published:** 2022-09-10

**Authors:** Jurij Janež, Gašper Horvat, Aleš Jerin, Jan Grosek

**Affiliations:** 1Department of Abdominal Surgery, University Medical Centre Ljubljana, 1000 Ljubljana, Slovenia; 2Faculty of Medicine, University of Ljubljana, 1000 Ljubljana, Slovenia; 3Clinical Institute of Clinical Chemistry and Biochemistry, University Medical Centre Ljubljana, 1000 Ljubljana, Slovenia; 4Faculty of Pharmacy, University of Ljubljana, 1000 Ljubljana, Slovenia

**Keywords:** colorectal surgery, anastomosis, anastomotic leak, biochemical markers, lactate, carcinoembryonic antigen, peritoneal fluid

## Abstract

*Background and Objectives*: The aim of our study was to evaluate the value of leukocyte, C reactive protein (CRP), procalcitonin, lactate, and carcinoembryonic antigen (CEA) in blood and peritoneal fluid in early recognition of anastomotic leak (AL) after colorectal resections. *Materials and Methods*: Our pilot prospective cohort study was conducted at the abdominal surgery department at University Medical Center Ljubljana. A total of 43 patients who underwent open or laparoscopic colorectal resection because of benign or malignant etiology were enrolled. All of the patients had primary anastomosis without stoma formation. *Results*: Three patients in our patient group developed AL (7%). We found a statistically significant elevation of serum lactate levels in patients that developed AL compared to those who did not but noted no statistically relevant difference in the blood or peritoneal fluid levels of other biochemical markers. *Conclusions*: Elevated lactate levels may be considered a promising biomarker for the early diagnosis of AL, but more research on bigger patient groups is warranted.

## 1. Introduction

Colorectal cancer is one of the most common malignant diseases worldwide. The mainstay of treatment is surgical resection, taking into account the oncological principles of adequate resection margins and lymphadenectomy. Complete, margin-negative resection confers the greatest chance for a cure. Hence, all the resection margins must have no microscopic cancer cell residues, in addition to the removal of at least 12 lymph nodes for adequate staging. The most unwanted postoperative complication is an anastomotic leak (AL), which often requires a second operation, usually necessitating the formation of either a double-barrel or terminal stoma. The key contributing surgical factors for anastomotic leak are anastomotic technique, surgical approach, and duration of surgery [1,2]. Moreover, not all anastomotic leaks are recognized early enough; hence, many patients present with the signs of systemic inflammatory response syndrome (SIRS) and sepsis. The morbidity and mortality are increased in these patients, as are the local recurrence of malignant disease, length of hospital stay, and overall costs of treatment [3,4]. 

Some studies have evaluated different serum and peritoneal fluid biomarkers for early recognition of an AL, but no biomarker has proven sufficiently useful to be implemented in routine clinical practice [5,6]. Biomarkers of AL are usually divided into three categories: biomarkers of ischemia, inflammation, and tissue repair. All have been suggested as potential early indicators of pathophysiological processes impeding anastomotic healing [5,6]. Carcinoembryonic antigen (CEA) is a glycoprotein produced by enterocytes, which can be overproduced by gastrointestinal cancer cells, serving as a tumor marker for colorectal malignancies. Its role as a potential biomarker of AL was recently proposed [7]. Our study aimed to evaluate the value of leukocyte, C-reactive protein (CRP), procalcitonin (PCT), lactate, and CEA levels in the blood and peritoneal fluid in the early recognition of anastomotic leak following colorectal resections.

## 2. Materials and Methods

Our pilot prospective cohort study was conducted at the Abdominal Surgery Department and the Clinical Institute of Clinical Chemistry and Biochemistry at the University Medical Center Ljubljana, Slovenia. Fifty patients were consecutively enrolled. The inclusion criteria were malignant or benign colorectal pathology, intention for restorative bowel resection with anastomosis, and age over 18 years. Patients signed informed consent for participation in the study. Exclusion criteria included advanced metastatic disease (stage 4), chronic inflammatory bowel disease, active infections, antibiotic therapy, immunosuppressive therapy, congenital or acquired immunodeficiency diseases, or hematological diseases. Patients in whom an urgent operation was necessary were excluded from the study. All patients in whom a stoma formation was intended were also excluded from the study. The study was reviewed and approved by the Republic of Slovenia National Medical Ethics Committee (Ethics approval number: 0120-476/2019/12).

Blood samples were taken from all patients on the day before surgery. EDTA blood concentrations of leukocytes, CRP, PCT, lactate, and CEA were measured. The same procedure was repeated 6–8 h after the surgery and every day from the first to the fifth postoperative day. At the same time intervals after the surgery, the abdominal drain fluid was collected for measurement of leukocytes, lactate, and CEA concentrations in the abdominal drain fluid.

The patients’ demographic data were collected, and postoperative complications were evaluated. Anastomotic leak was considered along with all conditions with clinical or radiological features of anastomotic dehiscence. Hence, it was defined, as per the International Study Group definition, as a confirmed defect of the intestinal wall at the anastomotic site (including suture and staple lines) leading to a communication between the intra- and extraluminal compartments. Severity of anastomotic leakage should be graded according to the impact on clinical management. Grade A anastomotic leakage results in no change in patients’ management, whereas grade B leakage requires active therapeutic intervention but is manageable without re-laparotomy. Grade C anastomotic leakage requires re-laparotomy [8]. Patients who suffered severe infective complications after surgery (not connected with AL), such as pneumonia, urinary tract infection, or surgical wound site infection, and needed antibiotic therapy were excluded from the study and further analysis. Our perioperative antibiotic protocol for colorectal resections is one dose of gentamicin and metronidazole prior to surgery (30–60 min before skin incision), then two more doses of metronidazole every 8 h. If there was contamination during the procedure, this antibiotic protocol is extended to three days after surgery. In our institution, we apply mechanical bowel preparation before surgery. We do not use oral antibiotics, as in some other institutions [9].

Continuous variables were described as appropriate by means and standard deviations or medians and interquartile ranges. Categorical variables were described by frequencies and percentages. Continuous variables by study group were graphically presented with boxplots. Where there were extreme maximum values, the maximum value was written on the top of the boxplot. Due to the low number of patients with anastomotic leakage, the study was underpowered to detect differences between patients with and without anastomotic leakage. Nonetheless, the differences between the study groups in biomarker values were tested by the Mann–Whitney U test and treated as statistically significant when *p* < 0.005. The results, however, should be interpreted with caution. 

## 3. Results

### 3.1. Sample Description

A total of 43 patients were included in the study, 20 (46.5%) of which were males (Table 1). The mean age (SD) of the patients was 69.9 years (13.6). Three patients (7%) had anastomotic leakage (AL): one patient on the fifth, another on the sixth, and the last on the ninth postoperative day. The comparison of patients who experienced anastomotic leakage and others revealed no statistically significant differences regarding age (*p* = 0.537) or sex (*p* = 0.466). The median values of preoperative serum biomarkers with interquartile ranges are summarized in Table 1.

Of the forty-three patients, three (7%) had a preoperative diagnosis of benign disease. One of these patients underwent open resection of the colon, while the other two had a laparoscopic operation. The other forty patients (93%) were operated on because of adenoma or adenocarcinoma of the colon or rectum (Table 2). Thirteen patients underwent open resection, while twenty-seven underwent laparoscopic resection. The three patients that developed AL underwent distinct surgical procedures: open resection of the sigmoid colon, laparoscopic low anterior resection of the rectum, and laparoscopic resection of the sigmoid colon. All three were operated on because of a malignant disease, and all required reoperation. Three patients in whom AL occurred had some comorbidities. Two patients had arterial hypertension, diabetes mellitus type 2 on oral antidiabetic therapy, and hyperlipidemia. One patient had chronic obstructive pulmonary disease (COPD) and arterial hypertension. All three patients had stage III adenocarcinoma on final pathohistological examination.

### 3.2. Serum and EDTA Blood Biomarkers

The values of each biomarker by anastomotic leakage at each time point are illustrated in Figure 1, Figure 2, Figure 3, Figure 4 and Figure 5. At each time point, the differences were tested by the Mann–Whitney U test. No statistically significant differences in leukocyte count between the study groups were found at any time point, although there appeared to be a steeper increase in leukocyte count in the anastomotic leakage group one day (t1) after the operation (Figure 1).

The two groups were comparable in CEA levels at baseline (before operation), and no statistically significant differences were found between the study groups at any time point. The level of CEA appeared higher in the anastomotic leakage group on the third day after the operation (Figure 2).

The two groups were comparable in CRP values throughout the study (Figure 3). A slightly lower increase in CRP was apparent in the anastomotic leakage group one day after the operation.

Although the study groups were comparable in lactate levels at baseline (before the operation), a statistically significant difference between groups was found 6 to 8 h after the operation, with the anastomotic leakage group exhibiting higher lactate values (Figure 4). No statistically significant differences in lactate values between the groups were found on the first to the fifth day of operation.

The two groups were comparable in PCT values throughout the study, although somewhat higher levels were apparent in the anastomotic leakage group on the third postoperative day (Figure 5).

A higher increase in leukocyte count (*p* = 0.071) and lactate levels (*p* = 0.009) from baseline to 6–8 h after the operation was apparent in the anastomotic leakage group (Table 3). One day after the operation, the increase in CRP values appeared lower in the anastomotic leakage group (*p* = 0.06).

### 3.3. Abdominal Drainage Fluid Biomarkers

The two groups were comparable in leukocyte count in abdominal drainage fluid throughout the study (Figure 6).

Although the values of CEA 6–8 h after the operation were higher in the anastomotic leakage group, the difference was not statistically significant (Figure 7). Similarly, no statistically significant difference was found at the proceeding time points.

The lactate levels appeared slightly lower in the anastomotic leakage group on the first and the fifth day after the operation (Figure 8), but no statistically significant differences at any of the time points were found.

Median (IQR; *n*) biomarker values from serum by study group and Median (IQR; *n*) biomarker values from abdominal drainage by study group are presented in the Appendix A (Table A1 and Table A2).

## 4. Discussion

The present study aimed to evaluate the significance of several biomarkers for early recognition of leaking colorectal anastomosis. Our diverse patient group had three AL cases, representing an overall AL incidence of 6.9%. When calculating the incidence of AL in patients with a diagnosis of malignant disease, its value increases to 7.5%. Both numbers correlate well with the incidence of AL usually reported in benign or malignant colorectal surgery. When analyzing AL incidence based on the anatomical site of the anastomosis, we had 2 AL in the patient group that underwent sigmoid or rectosigmoid colon resection, representing a rate of 15.3%. Probably due to the small patient cohort, this was higher than the reported rate in the literature, which varies from 5.1% to 7.7% to as low as 1% [10,11,12]. One out of eight patients (or 12.5%) undergoing low anterior resection (LAR) for rectal cancer developed an AL, which is comparable to a reported AL rate of 13–19% after colorectal anastomosis [13,14]. 

In our group of patients, we did not find any statistically significant difference between the incidence of AL related to age or sex, which are both known independent risk factors for AL [14]. 

We found a statistically significant difference in serum lactate levels 6 to 8 h after surgery in patients with AL as opposed to patients that did not have anastomotic leakage. We did not find any significant difference between serum lactate levels at any other time point in the study. Lactate is an end product of anaerobic metabolism and therefore functions as one of the markers of ischemic metabolism [5]. With colonic wall ischemia at the site of the anastomosis as one of the most widely accepted mechanisms of AL pathogenesis, [15,16] the clinical significance of lactate levels (serum level, peritoneal fluid level, or lactate-pyruvate ratio in peritoneal fluid) has been widely investigated. In our study, serum lactate was elevated only in the immediate postoperative period in patients with AL; therefore, it could hardly be attributed to bowel wall ischemia as the cause of later AL. However, its role as an early predictor of AL, when analyzed in serum or peritoneal fluid, remains to be conclusively established. Nevertheless, promising results from previous studies justify further exploration of lactate peritoneal or serum levels as a possible early biomarker for AL [5,6,17]. 

In our study, we could not confirm the role of the peritoneal lactate level as an early biomarker of AL. This is partly due to the small population sample or other limitations related to sample collection. We could not guarantee that the intra-abdominal tip of the drain was placed in the imminent vicinity of the anastomosis permanently, as the drain tube was only secured at the skin level. Peristalsis, patient movement, or movement of the intraperitoneal fluid could all potentially displace a drain [6,18]. This problem was addressed in a human pilot study by Jansson et al., where dialysis microcatheters were used as more accurate measurement devices at the site of the anastomosis. They confirmed the role of raised peritoneal lactate/pyruvate (L/P) levels as an early predictive marker for AL, warranting further research [19,20]. Other investigators made similar findings, further establishing the role of lactate and L/P levels in the peritoneal fluid as a potential early biomarker for AL [21]. It has been shown that peritoneal biomarkers of AL have a potential for becoming standardized and routinely utilized biomarkers, but further studies with standardized sample collection and analysis on a larger patient population are crucially needed [17,22].

We did not find any statistical difference between CRP serum levels in patients with or without AL, which can probably be attributed to the relatively small patient cohort. The role of CRP as a reliable marker of inflammatory process is indisputable, but it lacks specificity and positive predictive value for diagnosing AL, as there are various reasons for raised CRP levels in the postoperative period [5,14]. On the other hand, it has been determined that it has an excellent negative predictive value for AL between the third and fifth postoperative days [23].

We also did not find any statistical difference between PCT levels at any time point in our study. Raised levels of PCT correspond only with bacterial infection and not with the inflammatory process of any other origin. Therefore, higher PCT values correspond to bacterial infection at the site of AL (or any other bacterial inflammation such as sepsis, pneumonia, urinary tract infection, etc.). PCT values show an almost immediate response to bacterial infection, as its raised levels can be seen 2–3 h after the release of bacterial endotoxins or other inflammatory cytokines. Its role as a predictive factor for AL has been explored in different studies with promising results, both in combination with CRP levels and as a standalone biomarker. Different cut-off levels at different postoperative days were proposed [5].

Previous studies on blood EDTA leukocyte levels did not find it a useful biomarker of AL, as it lacks specificity and sensitivity [5]. Similarly, we did not find any statistically significant difference between our patient groups regarding leukocyte levels in the serum and the peritoneal fluid.

In a recent study by Berkovich et al., a role of CEA as an early biomarker of AL in peritoneal drainage fluid was proposed, with a significant elevation recorded as soon as 6 h after surgery. When comparing serum CEA levels, they did not find any statistically significant difference [7]. Our study found no statistically important differences between serum and peritoneal levels of CEA in patients that developed AL. CEA is a newly proposed biomarker of AL, with our study being the only one in addition to the original proposers to address its usefulness in patients after colorectal surgery. Further research in this regard is therefore warranted. There are some inflammatory and blood cell count indexes that have been recently correlated to AL after colorectal surgery. Among them are neutrophil-to-lymphocyte ratio (NLR), platelet-to-lymphocyte ratio (PLR), and CRP-to-albumin ratio (CAR) [24]. Currently, none of them proved to be specific or sensitive enough for diagnosing AL after colorectal surgery [24,25]. Additionally, further studies are needed for elucidating the potential role of them in diagnosing AL after colorectal surgery.

Our study had some limitations, the foremost being the small number of patients included. We also had a very heterogeneous patient group: benign and malignant operative indication, different types of bowel anastomosis, and different surgical approaches. The heterogenicity is similarly evident in other studies, given the obvious difficulties in enrolling a sufficiently large number of patients that were operated on in a single-center study. Some of the methods for early prediction of AL are also expensive and require more technical expertise, such as, for example, the use of microdialysis catheters. Implementation of their use in day-to-day practice is somewhat more difficult and expensive. They were therefore not included in our and many other studies. 

## 5. Conclusions

We found some promising results regarding serum lactate levels as an early postoperative marker of AL. Nevertheless, the search for a definitive early biomarker of AL continues given the lack of conclusive evidence on any blood or peritoneal biomarkers. Further prospective studies on larger patient populations are still needed to establish definitive biomarkers that could be useful for the early recognition of AL. 

## Figures and Tables

**Figure 1 medicina-58-01253-f001:**
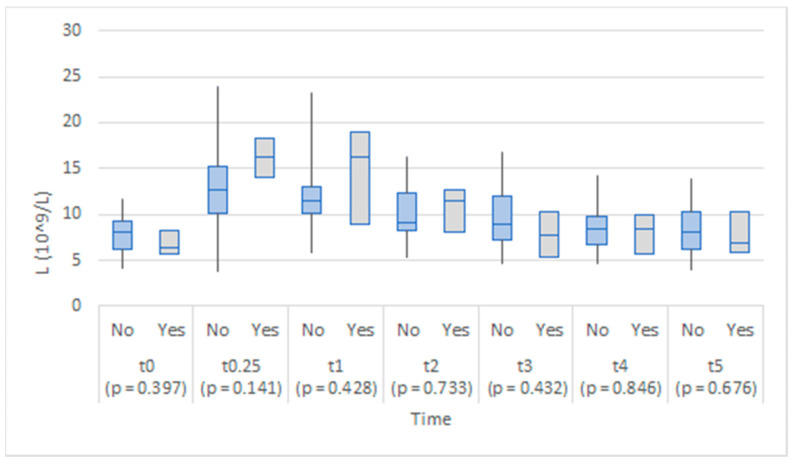
Box plots of leukocyte count in EDTA blood at each time point by anastomotic leakage (No/Yes).

**Figure 2 medicina-58-01253-f002:**
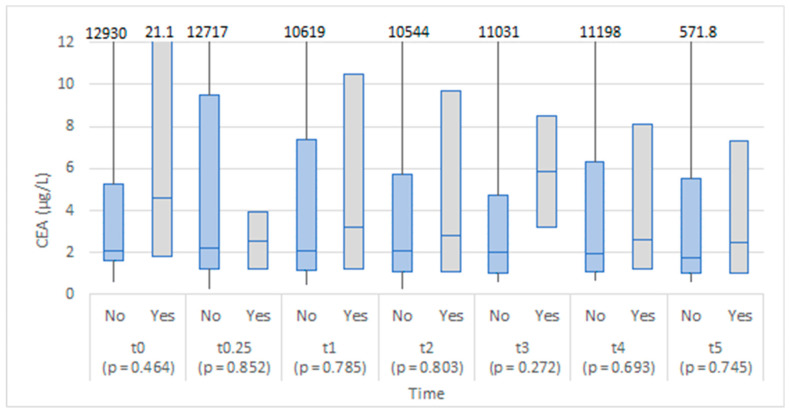
Box plots of CEA in serum at each time point by anastomotic leakage (No/Yes).

**Figure 3 medicina-58-01253-f003:**
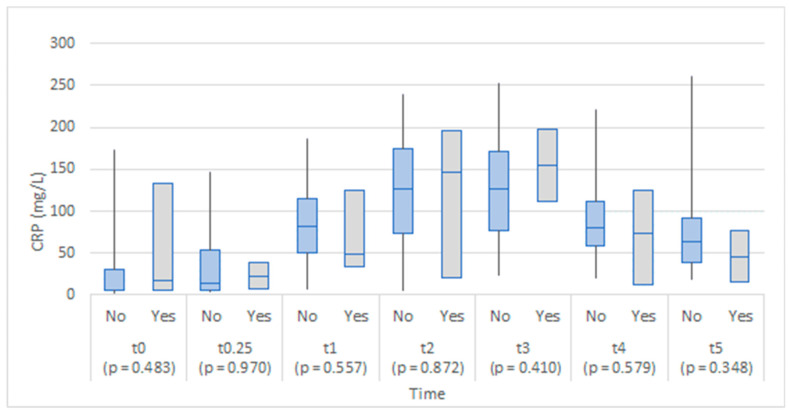
Box plots of CRP in serum at each time point by anastomotic leakage (No/Yes).

**Figure 4 medicina-58-01253-f004:**
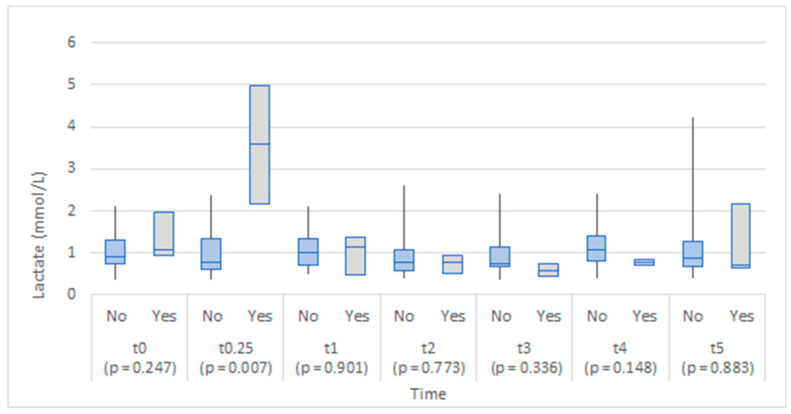
Box plots of lactate in serum at each time point by anastomotic leakage (No/Yes).

**Figure 5 medicina-58-01253-f005:**
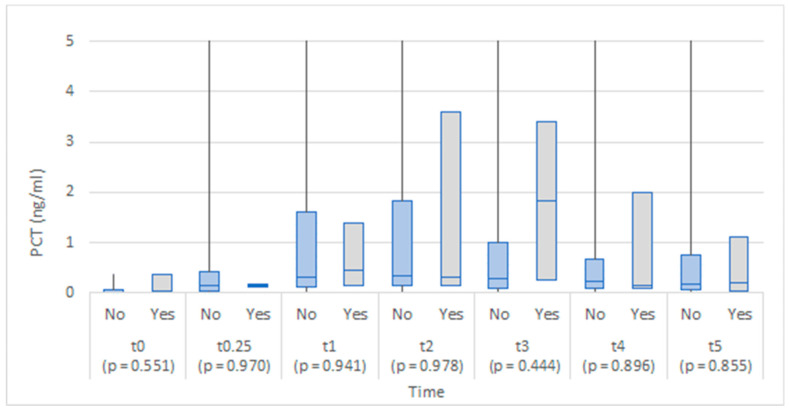
Box plots of PCT in serum at each time point by anastomotic leakage (No/Yes).

**Figure 6 medicina-58-01253-f006:**
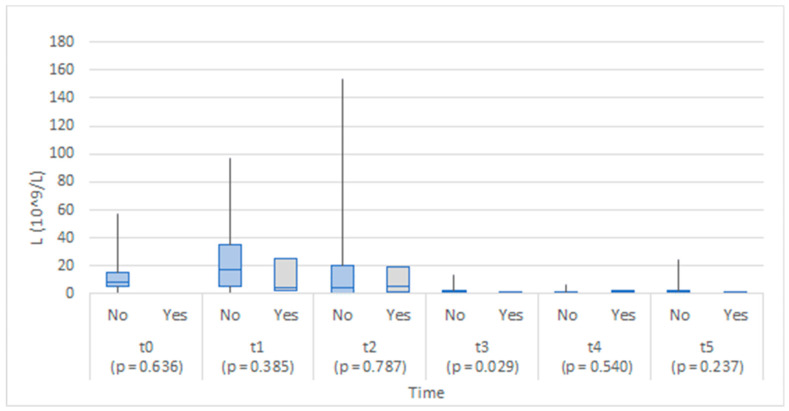
Box plots of leukocyte count in abdominal drainage fluid at each time point by anastomotic leakage (No/Yes).

**Figure 7 medicina-58-01253-f007:**
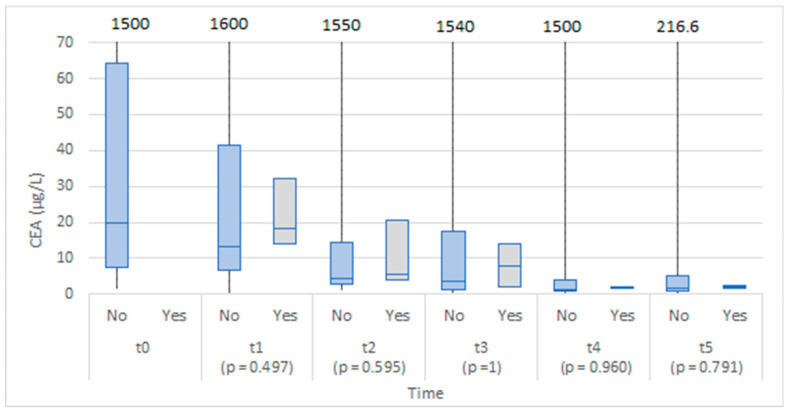
Box plots of CEA in abdominal drain fluid at each time point by anastomotic leakage (No/Yes).

**Figure 8 medicina-58-01253-f008:**
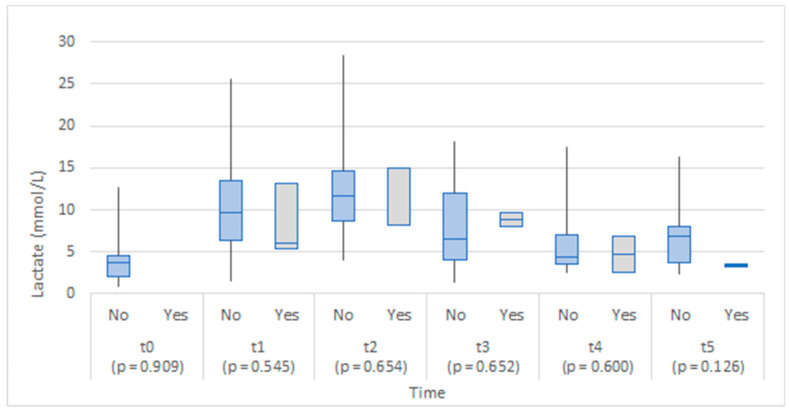
Box plots of lactate in abdominal drainage fluid at each time point by anastomotic leakage (No/Yes).

**Table 1 medicina-58-01253-t001:** Patient characteristics.

	***n* = 43**
Mean age (SD)	69.9 (13.6)
Male sex *n* (%)	20 (46.5)
Anastomotic leakage *n* (%)	3 (7)
Serum biomarkers before operation	
Me (IQR) L	8 (6.2–9.3)
Me (IQR) CRP	6 (5–31)
Me (IQR) PCT	0.02 (0.02–0.06)
Me (IQR) lactate	0.96 (0.74–1.3)
Me (IQR) CEA	2.2 (1.7–5.8)

**Table 2 medicina-58-01253-t002:** Type of surgery.

Type of Surgery	Number of Patients (%)
**Laparoscopic** Right hemicolectomy Resection of the sigmoid colon LAR ^1^ Resection of the rectosigmoid Resection of the transverse colon	**29 (67.4%)** 14 (48.3%) 7 (24.1%) 5 (17.2%) 2 (6.9%) 1 (3.4%)
**Open** Right hemicolectomy Resection of the sigmoid colon LAR ^1^ Resection of the rectosigmoid Resection of the transverse colon	**14 (32.6%)** 7 (50%) 2(14.3%) 3 (21.4%) 2 (14.3%) 0

^1^ Low anterior rectum resection.

**Table 3 medicina-58-01253-t003:** Change in serum biomarkers from baseline (t0) to 6–8 h after the operation (t1) to 1 day after the operation (t2).

	Anastomotic Leakage		*p*
	No	Yes	
L ∆t1 − t0	4.7 (2.25–7.25; 32)	8.85 (7.7–10; 2)	0.071
CRP ∆∆t1 − t0	0 (−1–5; 30)	−46.5 (−95–2; 2)	0.403
PCT ∆∆t1 − t0	0.13 (0.01–0.32; 29)	0.12 (0.09–0.15; 2)	0.903
Lactate ∆∆t1 − t0	0.02 (−0.36–0.5; 28)	2.58 (1.1–4.05; 2)	0.009
CEA ∆∆t1 − t0	−0.4 (−0.7–0; 29)	−0.65 (−0.7–−0.6; 2)	0.348
L ∆t2 − t0	3.4 (1.7–5.8; 37)	7.9 (3.3–12.6; 3)	0.104
CRP ∆t2 − t0	57 (31.5–81.5; 36)	28 (−8–32; 3)	0.06
PCT ∆t2 − t0	0.3 (0.12–1.44; 34)	0.41 (0.13–1.01; 3)	0.979
Lactate ∆t2 − t0	0.09 (−0.32–0.45; 31)	0.06 (−1.49–0.45; 3)	0.688
CEA ∆t2 − t0	−0.55 (−1.15–−0.2; 32)	−1.4 (−10.6–−0.6; 3)	0.14

## Data Availability

Not applicable.

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
