# Peer review of "The Significance of Blood and Peritoneal Fluid Biochemical Markers in Identifying Early Anastomotic Leak following Colorectal Resection—Findings from a Single-Center Study"

_medicina, 2022, doi:10.3390/medicina58091253_

Round 1

Reviewer 1 Report

This study aims evaluating the ability of several biomarkers (leukocytes, CRP, PCT, lactate, and CEA) to predict early anastomotic leak (AL) in a series of 43 patients who underwent colorectal resection. It is a prospective series with a very limited number of subjects, and patients seems quite heterogenous (benign/malignant diagnoses, laparoscopic/open procedures, right/left colectomies and rectum resections). The Authors concluded that increased serum lactate levels may be considered a promising biomarker for early detection of AL. The manuscript is well written, however results are not supported by adequate evidence. 

I have the following remarks:

·  In the Introduction (line 39) the sentence “Some studies have evaluated different serum and peritoneal fluid biomarkers…” should be supported with some references.

·  In the inclusion/exclusion criteria (lines 53-60) it should be specified if urgent operations were included or not.

·  Ethics approval number/ID should be included in the Materials and Methods (line 61).

·  Patient comorbidities and advanced disease stage may favor postoperative complications, and these factors should be reported for subject with and without AL.

·  There are several inflammatory indexes (e.g. neutrophil-to-lymphocyte ratio, platelet-to-lymphocyte ratio, CRP to albumin ratio) that have been recently correlated to AL after colorectal surgery. This issue should be considered in the discussion. 

·  Serum lactate levels were significantly increased in the immediate postoperative period in patients with AL, however this was not confirmed on the first to the fifth PO day. These results are rather difficult to interpret if the elevated serum lactate is related to bowel wall ischemia as postulated in the Discussion (lines 204-208).

Author Response

Dear Reviewer,

the answers to your commnets are in the attached file.

Sincerely, Jurij Janež

Reviewer 2 Report

Dear Authors, 

I had the pleasure to review "The significance of blood and peritoneal fluid biochemical markers in identifying early anastomotic leak following colorectal resection – findings from a single-center study"

The idea was not new but was not researched enough so i congratulate you for the decision and the work.

here are my comments:

Bipolar stoma – try other name line 34

Methods section

Line 69-73 there is inaccurate classification of the leakage, especially not accordingly with the citated paper [6] – they have three grades of leakage A, B, C, you have something unclear. You may stay with your classification and remove the citation, or you should redo the work according with this famous paper.

Line 74-75 – organ space infection as a complication post-surgery involved a kind of leakage in many cases, so be more specific about these infections. You don’t mentioned the antibiotics protocol – one dose during surgery, repeated dose if the surgery was longer than 2 hours, or antibiotics if there was a contamination during the procedure.

Line 81-82 The study was underpowered – by how many patients?

Results – what happened with 7 patients – reason for exclusion, maybe a flow-chart

The number was very low – you should consider continuing the study only of rectal resection and adding numbers.

Also there are other causes of possible error for lactate increased which you may consider in future research.

Author Response

Dear Reviewer,

the answers to your comments are in the attached file.

Sincerely,

Jurij Janež

Round 2

Reviewer 1 Report

The Authors have satisfactorily responded to all my comments and made the necessary changes to the manuscript.